# Unveiling a Shift in the Rotavirus Strains in Benin: Emergence of Reassortment Intergenogroup and Equine-like G3P[8] Strains in the Post-Vaccination Era

**DOI:** 10.3390/v17081091

**Published:** 2025-08-07

**Authors:** Jijoho M. Agbla, Milton T. Mogotsi, Alban G. Zohoun, Nkosazana D. Shange, Annick Capochichi, Ayodeji E. Ogunbayo, Rolande Assogba, Shainey Khakha, Aristide Sossou, Hlengiwe Sondlane, Jason M. Mwenda, Mathew D. Esona, Martin M. Nyaga

**Affiliations:** 1National Public Health Laboratory, Ministry of Public Health, Cotonou 01 P.O. Box 418, Benin; comlanz@yahoo.fr (A.G.Z.); assogbarolande@gmail.com (R.A.); 2Next Generation Sequencing Unit, School of Biomedical Sciences, Faculty of Health Sciences, Bloemfontein 9300, South Africa; mogotsimt@ufs.ac.za (M.T.M.); shangend@ufs.ac.za (N.D.S.); ogunbayoae@ufs.ac.za (A.E.O.); sondlaneh@ufs.ac.za (H.S.); nyagamm@ufs.ac.za (M.M.N.); 3Division of Virology, School of Pathology, Faculty of Health Sciences, University of the Free State, Bloemfontein 9300, South Africa; 4Faculty of Health Sciences, University of Abomey-Calavi, Cotonou 01 P.O. Box 526, Benin; 5Centre Hospitalier Universitaire de Zone (CHUZ)—Suru Léré Hospital, Ministry of Public Health, Cotonou 01 P.O. Box 418, Benin; marwar68@yahoo.fr; 6Wellcome Trust Research Laboratory, Christian Medical College and Hospital Vellore, Tamil Nadu 632004, India; shainey.rnc0411@gmail.com; 7World Health Organization-Benin (WCO-Benin), Cotonou 01 P.O. Box 918, Benin; sossoua@who.int; 8Diarrhea Pathogens Research Unit, Department of Virology, Sefako Makgatho Health Sciences University, Pretoria 0208, South Africa; jason.mwenda23@gmail.com (J.M.M.); mathew.esona@gmail.com (M.D.E.)

**Keywords:** rotavirus, equine-like G3P[8], genomic surveillance, reassortment, intergenogroup, Benin

## Abstract

While a global downward trend in rotavirus diarrhea cases has been observed following vaccine introduction, reassortment, genetic drift, and vaccine-escaping strains remain a concern, particularly in Sub-Saharan Africa. Here, we provide genomic insights into three equine-like G3P[8] rotavirus strains detected in Benin during the post-vaccine era. Whole-genome sequencing was performed using the Illumina MiSeq platform, and genomic analysis was conducted using bioinformatics tools. The G3 of the study strains clustered within the recently described lineage IX, alongside the human-derived equine-like strain D388. The P[8] is grouped within the lineage III, along with cognate strains from the GenBank database. Both the structural and non-structural gene segments of these study strains exhibited genetic diversity, highlighting the ongoing evolution of circulating strains. Notably, we identified a novel NSP2 lineage, designated NSP2-lineage VI. Amino acid comparisons of the G3 gene showed two conservative substitutions at positions 156 (A156V) and 260 (I260V) and one radical substitution at position 250 (K250E) relative to the prototype equine-like strain D388, the equine strain Erv105, and other non-equine-like strains. In the P[8] gene, three conservative (N195G, N195D, N113D) and one radical (D133N) substitutions were observed when compared with vaccine strains Rotarix and RotaTeq. These findings suggest continuous viral evolution, potentially driven by vaccine pressure. Ongoing genomic surveillance is essential to monitor genotype shifts as part of the efforts to evaluate the impact of emerging strains and to assess vaccine effectiveness in Sub-Saharan Africa.

## 1. Introduction

Rotavirus group A (RVA) remains a major cause of diarrhea among children under five years of age worldwide, with Sub-Saharan Africa most affected. The global mortality estimate attributed to rotavirus is approximately 128,500 deaths, with the majority (104,733) occurring in Sub-Saharan Africa [1]. The predominant circulating strains, which represent more than 90% of those detected globally, are G1, G3, G4, G9, and G12 in combination with P[8], alongside G2P[4], G2/G3/G12P[6], and some less common genotypes, such as G8P[4] and G6P[4]. The former group is typically associated with a Wa-like genetic backbone, while the latter is characterized by a DS-1-like genetic constellation [2,3,4,5,6,7,8].

Rotavirus contains eleven genome segments encoding six viral structural proteins (VP1-VP4, VP6, and VP7) and five or six non-structural proteins (NSP1-NSP5/6), with the VP7 and VP4 proteins determining the G and P antigenic specificities that define the virus’s binomial nomenclature as GxPx (x = genotype) [9]. Since 2008, to enhance the study of reassortment events, point mutations, interspecies transmission, genetic drift, and shifts within the segmented rotavirus genome, a comprehensive nomenclature has been adopted that considers all 11 gene segments, with VP7, VP4, VP6, VP1, VP2, VP3, NSP1, NSP2, NSP3, NSP4, and NSP5 denoted by the genotypes Gx, P[x], Ix, Rx, Cx, Mx, Ax, Nx, Tx, Ex, and Hx, respectively. This system has enabled the classification of three primary genotypic constellation backbones: Wa-like (I1-R1-C1-M1-A1-N1-T1-E1-H1; porcine origin), DS-1-like (I2-R2-C2-M2-A2-N2-T2-E2-H2; bovine origin), and AU-1-like (I3-R3-C3-M3-A3-N3-T3-E3-H3; feline origin) [5,10,11]. At least 42 G and 58 P genotypes have been described so far [12] (http://rega.kuleuven.be/cev/viralmetagenomics/virus-classification, accessed on 11 July 2024).

As per the World Health Organization (WHO) recommendations in 2009 [13,14,15], hitherto, four vaccines have since been introduced to support public health initiatives for controlling rotavirus diarrhea. These include Rotarix^®^ (RV1), a human monovalent vaccine with a G1P[8] strain (GlaxoSmithKline Biologics, Rixensart, Belgium); RotaTeq^®^ (RV5), a pentavalent bovine-human reassortant vaccine incorporating G1P[5], G2P[5], G3P[5], G4P[5], and G6P[8] strains (Merck Vaccines, Whitehouse Station, NJ, USA); ROTAVAC^®^, a monovalent vaccine with a G9P[11] strain (Bharat Biotech, Hyderabad, India); and Rotasiil^®^, a lyophilized pentavalent human-bovine reassortant vaccine containing the G1, G2, G3, G4, and G9 genotypes (Serum Institute of India, Pune, India) [16,17,18,19,20,21].

In Benin, the ROTAVAC^®^ vaccine was introduced into the national Expanded Program on Immunization (EPI) in December 2019 [8]. After the introduction of the vaccine, a downward trend in rotavirus prevalence was observed, with rates decreasing from 40.0% to 25.0% (unpublished surveillance data from Benin). This notable decline has also unveiled an unforeseen shift in the enteric virus ecology, marked by the emergence of equine-like G3P[8] strains, distinct from the classical Wa-like G3P[8], and bearing a DS-1-like genotype constellation, along with other novel genotypes such as G8P[4] and G8P[6].

The G3P[8], I2, R2, C2, M2, A2, N2, T2, E2, and H2 strains emerged in East Asia (Thailand, Indonesia) and Australia in 2013 and subsequently expanded into Europe (Germany, Spain), South America (Argentina), and the USA in the following years [22,23,24,25,26,27,28]. In Africa, Mozambique reported some G3P[8] strains that clustered very differently from the global equine-like G3 strains [29]. To date, only Kenya and Tanzania have published a report on equine-like G3P[8] strains [30,31]. This marks the first occurrence of this genotype in Benin, which appears to have undergone a complete intergenogroup reassortment event, incorporating an equine-like G3 VP7, a P[8] VP4, and a DS-1-like genetic backbone [32,33].

Severe diarrheal episodes in children have been reported in several countries, linked to the emergence of equine-like G3P[8] strains [23,24,34]. Given that RVA is largely host-specific, the detection of strains with an equine origin has underscored the importance of this study. The primary objective of this study is to conduct a comprehensive analysis of three equine-like strains identified in Benin in the post-vaccination period and to evaluate the influence of vaccine pressure on these strains. We sought to elucidate, within the country-specific context, the mechanisms by which novel viral variants in children propagate within the broader population. The insights gained will inform targeted intervention strategies and provide critical evidence for the need to enhance and sustain enteric virus surveillance in Benin.

## 2. Materials and Methods

### 2.1. Sampling Method

As part of the ongoing rotavirus surveillance system, stool samples are routinely collected from children under five years old with acute diarrheal diseases admitted to Suru Léré Teaching Hospital as per WHO rotavirus surveillance protocol. All stool samples were first screened at the National Public Health Laboratory in Benin with an enzyme immunoassay technique using the ProSpecT Rotavirus kit (ProSpecT™; Oxoid, Basingstoke, UK). Thereafter, a total of 89 rotavirus-positive stool samples were shipped under a Material Transfer Agreement (MTA), in dry ice, to the University of the Free State—Next Generation Sequencing (UFS-NGS) Unit, a WHO Collaborating Centre for Vaccine Preventable Disease (VPD) Surveillance and Pathogen Genomics in South Africa, for sequencing. Of the 89 rotavirus-positive samples, 86 (96.6%) were successfully whole-genome sequenced, with three samples identified as carrying the equine-like G3 genotype.

The three stool samples reported here were collected during the post-vaccine period from 2020 to 2022.

The first sample, identified as RVA/Human-wt/BEN/20P093/2020/G3P[8], was obtained from a 5-month-old male infant residing in Sémè-Kpodji. The patient presented with a 5-day history of diarrhea without vomiting and exhibited a relatively mild clinical presentation (no signs of lethargy or unconsciousness, no sunken eyes, and normal drinking behavior), although moderate dehydration was noted. The second sample, RVA/Human-wt/BEN/22011/2022/G3P[8], was collected from an 8-month-old male infant from Cotonou. The child presented with fever (38.3 °C), diarrhea, and vomiting (4 episodes per day), without lethargy or unconsciousness, and maintained normal drinking behavior. However, sunken eyes and moderate dehydration were observed. The third sample, RVA/Human-wt/BEN/22013/2022/G3P[8], was derived from a 9-month-old male infant, also from Sémè-Kpodji. This patient presented with diarrhea and vomiting (4 episodes per day), accompanied by lethargy/unconsciousness, sunken eyes, poor drinking behavior, and moderate dehydration, but no fever was reported.

All three children had received the ROTAVAC^®^ vaccine as part of the national immunization program.

### 2.2. Extraction of Nucleic Material and Purification

Nucleic material was extracted from 10% stool samples (100 µL or 1 g or a peanut-sized stool sample in 900 µL of phosphate-buffered saline (PBS) using an automated viral nucleic material extraction protocol on Chemagic^TM^ 360 (Mustionkatu 6, Turku, Finland) as per the manufacturer’s instructions. The extracted material was purified using the Qiagen MinElute gel extraction kit (Qiagen, Hilden, Germany). The purified nucleic material has then undergone a 1% 0.5× TBE agarose gel electrophoresis (Bioline, London, UK) stained with Pronasafe (Condalab, Madrid, Spain) at 100 volts for 40 min to assess the integrity of the nucleic material.

### 2.3. Complementary DNA (cDNA) Synthesis and PCR

The cDNA synthesis has been performed using a modified version of the Maxima H Minus Double-Stranded cDNA Synthesis kit and protocol (ThermoFisher Scientific, Waltham, MA, USA) as previously described by Mwangi and colleagues [35].

Briefly, the first strand cDNA synthesis was performed by denaturing a volume of 13 µL of extracted dsRNA product at 95 °C for 5 min. A volume of 1 µL of random hexamer primer was then added to the ssRNA obtained from the previous step and incubated at 65 °C for 5 min. The 4× First Strand Reaction Mix and the First Strand Enzyme Mix were added to the reaction tube for a final volume of 20 µL. The mixture was placed in the thermocycler set to 25 °C for 10 min, followed by 50 °C for 120 min, and 85 °C for 5 min. The first strand cDNA was further used as a template to synthesize the second strand cDNA by adding to the above reaction tube (20 µL), 5× Second Strand Reaction Mix, the Second Strand Enzyme, and nuclease-free water for a total reaction volume of 100 µL. The reaction mixture was incubated at 16 °C for 60 min, and subsequently, a volume of 6 µL of 0.5 M EDTA, pH 8.0, and 10 µL of the RNase were added to terminate the reaction and remove all residual RNA, respectively. Finally, the tube was kept at –20 °C until the purification step. The cDNA was then purified using the MSB^®^ Spin PCRapace kit (Stratec Molecular, Berlin, Germany) following the manufacturer’s instructions and eluted in 12 µL of elution buffer.

### 2.4. Rotavirus Whole Genome Sequencing

The Nextera XT DNA Library Preparation Kit (Illumina, San Diego, CA, USA) was used to construct DNA libraries for rotavirus whole genome sequencing. First, purified rotavirus cDNA was subjected to tagmentation as follows: 10 µL of Tagment DNA buffer was added to each well of a PCR plate containing a 5 μL volume of purified cDNA, within the 0.2–0.3 ng/μL concentration range, followed by the addition of 5 μL of Amplicon Tagment Mix. The PCR plate was centrifuged at a force of 280× *g* for 1 min and subsequently incubated in a thermocycler at 55 °C for 5 min, with a holding temperature of 10 °C. A 5 μL volume of Neutralize Tagment buffer was added to terminate the transposase enzyme’s activity, and the plate was kept at room temperature for 5 min.

Subsequently, the index primers (10 µL) as well as Nextera PCR Mix (15 µL) were added to the enzymatically fragmented and tagged DNA, followed by the amplification step under the following PCR conditions: 72 °C for 3 min, then 12 cycles at 95 °C for 10 s, 55 °C for 30 s, and 72 °C for 30 s, and finally, at 72 °C for 5 min, with a 10 °C hold temperature. The resulting amplified and uniquely barcoded products (DNA library) were purified using AMPure XP beads (Beckman Coulter, Pasadena, CA, USA) and freshly prepared 80% ethanol. The DNA library quality was then assessed using an Agilent 2100 BioAnalyzer (Agilent Technologies, Waldbronn, Germany) and quantified on Qubit, after which the normalized DNA libraries were pooled into a single tube and denatured using freshly prepared 0.2 N sodium hydroxide (NaOH). Finally, the pooled DNA library was diluted with hybridization buffer, and a 5% PhiX sequencing control was added until a final concentration of 8 pM was reached. The 8 pM library was then sequenced on the Illumina MiSeq platform (Illumina, San Diego, CA, USA) for 600 cycles using a MiSeq reagent v3 kit (Illumina Inc., San Diego, CA, USA) to generate 301 bp x 2-paired-end reads.

### 2.5. Data Analysis

#### 2.5.1. Quality Control, Genome Assembly and Genotype Assignment

The raw sequencing data were submitted to the bioinformatics pipeline developed at the UFS-NGS Unit on the University of the Free State HPC server. Briefly, raw reads in FASTQ format were first subjected to quality control using the FastQC tool (version 0.12.1) [36] and summarized with MultiQC (version 1.14), on the HPC server [37]. Sequencing reads were then trimmed using the Fastp version 0.23.2 tool [38]. Genome assembly was performed using both de novo and reference-based approaches with SPAdes: genome assembler version 4.1.0 [39] and SAMtools version 1.21 Using HTSlib 1.21—© 2024 Genome Research Ltd [40] to obtain the full-length nucleotide sequence of each genome segment. The assembled genomes of the study strains were visualized using Artemis software Release 18.2.0 [41].

The resulting consensus sequences of all strains were used as query sequences to determine the genotypes of each of the 11 segments in the online database, Virus Pathogen Database and Analysis Resource (ViPR) [42].

#### 2.5.2. Sequence Alignments

Multiple sequence alignments of study strains with cognate sequences retrieved from the core_nt database on the NCBI website were performed using the MAFFT tool embedded in Jalview software version 2.11.4.1 [43] and further refined using MUSCLE CLI (version 5.1.linux64) [44] or MAFFT CLI (version 7.520) tools [45] on the HPC server.

The maximum likelihood phylogenetic tree was constructed using IQ-TREE multicore version 2.2.2.3 [46] with 1000 bootstrap replications for branch support, following the selection of the optimal substitution model using the ModelTest CLI tool (version 0.1.7) [47] on the HPC server.

Based on the corrected Akaike Information Criterion (AICc), the models identified as the best fit for the sequences were TPM1uf+G4 (for VP6, VP7, NSP5/6), GTR+I+G4 (for VP2, VP3, NSP1, NSP2), TIM3+I+G4 (for VP1, VP4, NSP3), and TrN+G4 for NSP4.

Pairwise distance matrices for nucleotide and amino acid sequences were calculated using the *p*-distance algorithm in MEGA 11 [48].

#### 2.5.3. VP7 and VP4 Antigenic Regions

The sequences of the study strains, along with the prototype equine-like strain D388, selected non-equine-like strains from Benin and Mozambique, and the equine strain, were aligned using the MAFFT algorithm embedded in the Jalview software suite [43]. The Jalview translation tool was utilized to derive the corresponding amino acid sequences from the nucleotide templates. Amino acid variations were systematically analyzed within antigenic regions that have been previously characterized in the literature [25,49].

## 3. Results

### 3.1. Metrics on the Study Strains

Metrics on the study strains were generated using the Seqkit CLI tool (version 2.10.0) [46] on contigs obtained from de novo assembly performed with SPAdes [36], along with reports generated after sequence submission to the Virus Pathogen Database and Analysis Resource (ViPR) [39] (Appendix A).

### 3.2. Phylogenetic Analysis

The ORF lengths for each gene segment were determined as follows: VP1 (3264 bp), VP2 (2637 bp), VP3 (2505 bp), VP4 (2325 bp), NSP1 (1461 bp), VP6 (1191 bp), NSP2 (951 bp), NSP3 (930 bp), VP7 (978 bp), NSP4 (525 bp), and NSP5 (600 bp). All three sequenced strains exhibited a DS-1-like G3P[8]-I2-R2-C2-M2-A2-N2-T2-E2-H2 genotype constellation.

#### 3.2.1. Phylogenetic Analysis of the Equine-like G3

Since the emergence of equine-like G3 strains, a new lineage, lineage-IX, has been proposed, and, to date, nine lineages have been identified (Figure 1) [25]. All study strains were classified within lineage IX, exhibiting a high degree of nucleotide (99.6–100%) and amino acid (99.7–100%) similarity (Table 1). The study strains formed clusters with all related equine-like G3P[8] strains, including the prototype strain RVA/Human-wt/AUS/D388/2013/G3P[8], and shared nucleotide similarities ranging from 99.0% to 99.2% and amino acid similarities between 99.1% and 99.4%. Phylogenetically, the study strains were most closely related and clustered to strains from Indonesia (RVA/Human-wt/IDN/STM50/2015/G3P[8], RVA/Human-wt/IDN/STM4/2015/G3P[8]), Slovakia (RVA/Human-wt/SVK/2771/G3, RVA/Human-wt/SVK/2490/G3), the Dominican Republic (RVA/Human-wt/DOM/3000503701/2014/G3P[8]), and Japan (RVA/Human-wt/JPN/IW16-10/2016/G3P[8]), with the highest nucleotide similarity of 99.2% and amino acid similarity of 99.7%. Notably, the Benin strains shared nucleotide similarities of 90.6–90.7% and amino acid similarities of 99.2–99.5% with the equine strain RVA/Horse-wt/IND/Erv105/2003-2005/G3P[X] (Table 1).

#### 3.2.2. Phylogenetic Analysis of P[8]

Four well-characterized lineages have been reported in the phylogenetic analysis of P8 genotypes [8,32,33,50]. The study strains were grouped within lineage III, exhibiting high nucleotide and amino acid similarities of 99.7–100% among themselves. The overall nucleotide and amino acid similarities between the study strains and cognate P[8]-lineage III strains from the GenBank database ranged from 93.3% to 100% and from 94.1% to 100%, respectively. Study strains were specifically grouped within sub-lineage III-a, which bifurcated into two major clades. The strain RVA/Human-wt/BEN/20P093/2020/G3P[8], isolated in 2020, showed closer phylogenetic relationships to strains associated with G3 genotypes, while the other two strains (RVA/Human-wt/BEN/22011/2022/G3P[8] and RVA/Human-wt/BEN/22013/2022/G3P[8]) clustered with strains displaying a G8 genotype.

RVA/Human-wt/BEN/20P093/2020/G3P[8] clustered with strains identified in the Dominican Republic (RVA/Human-wt/DOM/3000503706/2014/G3P[8]) and the equine-like prototype strain RVA/Human-wt/AUS/D388/2013/G3P[8], sharing a maximum nucleotide similarity of 99.1% and an amino acid similarity of 99.0%. Furthermore, this strain clustered with other G3P[8] strains isolated from Germany, Spain, India, Thailand, and Taiwan (Figure 2 and Table 1).

The remaining strains (RVA/Human-wt/BEN/22011/2022/G3P[8] and RVA/Human-wt/BEN/22013/2022/G3P[8]), which predominantly clustered with strains possessing a G8 genotype, exhibited nucleotide and amino acid similarities of 99.5–99.7% with strains isolated in Japan in 2019 (RVA/Human-wt/JPN/IW19-03/2019/G8P[8] and RVA/Human-wt/JPN/TYMC19-19/2019/G8P[8]) and in the USA in 2017 (RVA/Human-wt/USA/3000804093/2017/G8P[8] and RVA/Human-wt/USA/3000804062/2017/G8P[8]). Overall, the current study strains displayed moderately high nucleotide (98.9%) and amino acid (98.8%) similarities compared to strains previously isolated from Benin during the pre-vaccination period, which carried the G1P[8] genotype.

#### 3.2.3. Phylogenetic Analysis of VP1-3 and VP6

VP1 gene

A total of fourteen lineages have been described for the VP1 protein [50,51]. All study strains were classified within lineage V, with two major sub-lineages, V-1 and V-2, being identified in this study. The study strains clustered within sub-lineage V-1 and were further subdivided into two principal clades, V-1a and V-1b (Appendix A and Table 1). These strains exhibited nucleotide and amino acid similarities of 96.6% to 99.5% and 98.9% to 99.9%, respectively, among themselves, and nucleotide and amino acid similarities ranging from 93.7% to 99.1% and 98.4% to 99.8%, respectively, with cognate strains retrieved from GenBank.

Within clade V-1a, the strains RVA/Human-wt/BEN/20P093/2020/G3P[8] and RVA/Human-wt/BEN/22011/2022/G3P[8] clustered predominantly with African strains from Ghana, Malawi, and Zambia, including those previously identified in Benin with the G2P[4] specificity. These strains shared nucleotide and amino acid similarities of 98.6% to 99.1% and 98.8% to 99.7%, respectively, with the highest similarities observed between the study strains and the Benin pre-vaccine G2P[4] strains.

Clade V-1b comprised the study strain RVA/Human-wt/BEN/22013/2022/G3P[8] and other closely related strains isolated in India and the USA, showing a maximum nucleotide similarity of 99.3% (with RVA/Human-wt/USA/3000558215/2016/G2P[8] and RVA/Human-wt/IND/TN030007/2016/G2P[4]) and an amino acid similarity of 99.5% (with RVA/Human-wt/USA/3000558215/2016/G2P[8], RVA/Human-wt/IND/TN030007/2016/G2P[4], and RVA/Human-wt/IND/TN010491/2017/G2P[4]).

All study strains exhibited distinct clustering from the prototype strain D388. Notably, a pre-vaccine strain isolated in Benin in 2016, of the G3P[6] genotype (RVA/Human-wt/BEN/3001607144/2016/G3P[6]), appears to delineate a novel clade within sub-lineage V-2, distinct from the D388 strain.

VP2 gene

All strains were categorized within lineage IV among the fourteen recognized lineages to date [32,50,51]. They exhibited nucleotide and amino acid similarities ranging from 98.7% to 100% and 99.8% to 100%, respectively, among themselves. These strains, along with the D388 strain, constitute clade IV-1a. Within this clade, strains isolated in 2022 (RVA/Human-wt/BEN/22011/2022/G3P[8] and RVA/Human-wt/BEN/22013/2022/G3P[8]) demonstrated distinct clustering from the strain isolated in 2020 (RVA/Human-wt/BEN/20P093/2020/G3P[8]).

It is noteworthy that the 2022 strains cluster with strains exhibiting the G8P[8] specificity, which have been recently identified in the USA and Japan. These strains display very high nucleotide (99.5% to 99.8%) and amino acid (99.8% to 100%) similarities. In contrast, the RVA/Human-wt/BEN/20P093/2020/G3P[8] strain is closely related to typically equine-like strains isolated in Japan, the Dominican Republic, and India, as well as to the D388 strain. Nucleotide similarities for this strain ranged from 99.0% to 99.5%, while amino acid similarities ranged from 99.5% to 99.8% (Appendix A and Table 1).

VP3 gene

The study strains were classified within lineage VII, specifically within sub-lineage VII-1, and exhibited nucleotide and amino acid similarities ranging from 99.5% to 100%. These strains demonstrated a close genetic relationship with previously identified strains from Benin bearing the G2P[4] genotype, as well as with other G2P[4] and G9P[4] strains globally. The nucleotide similarity among these strains ranged from 95.9% to 99.1%, while the amino acid similarity ranged from 97.7% to 99.6%. These strains were distinctly separated from those associated with sub-lineage VII-2, which includes caprine strains.

It is noteworthy that the equine-like prototype strain D388 is categorized within lineage V, alongside pre-vaccine G12P[6] strains from Benin and strains from Ghana, Malawi, Germany, Japan, and the USA (Appendix A and Table 1).

VP6 gene

Among the fifteen recognized lineages [35,51], the study strains were classified within lineage V. They formed a distinct clade, separate from a pre-vaccine G2P[4] strain isolated in 2016, which is categorized as sub-lineage V-2, with a bootstrap support value of 98%. The study strains demonstrated nucleotide and amino acid similarities ranging from 97.9% to 100% and 99.7% to 100%, respectively.

In alignment with prior observations for the VP2 gene, the 2022 strains exhibited a closer genetic relationship with recently identified G8P[8] strains from Japan. The highest nucleotide (99.7%) and amino acid (100%) similarities were observed with strains RVA/Human-wt/JPN/Tokyo17-20/2017/G8P[8], RVA/Human-wt/JPN/TYMC19-16/2019/G8P[8], RVA/Human-wt/JPN/TYMC19-13/2019/G8P[8], and RVA/Human-wt/JPN/NS19-17/2019/G8P[8].

The RVA/Human-wt/BEN/20P093/2020/G3P[8] strain clustered with strains from Kenya, Rwanda, Malawi, India, and China, exhibiting the highest nucleotide (100%) and amino acid (100%) similarities with strains from Malawi (RVA/Human-wt/MWI/BID2QF/2014/G2P[4]), Kenya (RVA/Human-wt/KEN/KLF0578/2012/G2P[4]), and India (RVA/Human-wt/IND/CMC-00020/2012/G2P[X], RVA/Human-wt/IND/CMC-00024/2012/G2P[X], RVA/Human-wt/IND/CM-0286/2013/G2P[4]). Additionally, the study strains showed nucleotide similarities of 98.0% to 98.4% and amino acid similarities of 99.2% to 99.5% with the equine-like prototype strain D388 (Appendix A and Table 1).

#### 3.2.4. Phylogenetic Analysis of NSP1-5

NSP1 gene

All study strains clustered with cognate sequences retrieved from GenBank within lineage IV, exhibiting nucleotide and amino acid similarities of 93.8–99.2% and 93.3–99.2%, respectively [35,51]. Notably, the strain RVA/Human-wt/BEN/22013/2022/G3P[8] clustered with well-characterized Benin G2P[4] strains, sharing nucleotide similarities of 98.0–99.2% and amino acid similarities of 97.7–99.0%. In contrast, the strains RVA/Human-wt/BEN/22011/2022/G3P[8] and RVA/Human-wt/BEN/20P093/2020/G3P[8] demonstrated a closer genetic relationship with strains isolated in China, Japan, and India. They displayed high nucleotide similarity (98.9–99.0%) and amino acid similarity (99.2%) with the Japanese strains RVA/Human-wt/JPN/To16-30/2016/G3P[8] and RVA/Human-wt/JPN/To16-28/2016/G3P[8], as well as the Indian strains RVA/Human-wt/IDN/STM050/2015/G3P[8] and RVA/Human-wt/IDN/STM004/2015/G3P[8], including the prototype strain D388 (Appendix A and Table 1).

NSP2 gene

Two of the study strains appear to define a new lineage, designated as lineage VI. Strains within this lineage exhibit high nucleotide similarities, ranging from 98.6% to 100%, and amino acid similarities between 99.7% and 100%. When compared with other recognized lineages, lineage VI shows the highest similarity with lineages I and V. Nucleotide similarities reached 88.1% with lineage I, represented by the strain RVA/Human-tc/USA/DS-1/1976/G2P[4], and 86.4% to 87.1% with lineage V (refer to Table 1). Amino acid similarities were 94.6% for lineage I and ranged from 94.3% to 95.6% for lineage V. The observed cut-off values in terms of genetic distances between these lineages (refer to the p-distance table) support the hypothesis that the study strains, along with other indicated strains in the phylogenetic tree, likely define a new lineage within the NSP2 gene, genogroup N2.

In contrast, the strain RVA/Human-wt/BEN/20P093/2020/G3P[8] forms a well-supported cluster with the Chinese strain Fuzhou 20-32, as well as strains primarily originating from Kenya (Appendix A). These strains display nucleotide similarities ranging from 98.5% to 99.1%, with amino acid similarities of 99.4% to 99.7% (Table 1).

NSP3 gene

Among the seven recognized lineages [35,51], the study strains were grouped within lineage V, forming distinct clades within this classification. The 2022 strains closely aligned with the G8P[8] strains predominantly isolated in the USA, Japan, Thailand, and Vietnam in 2019, as well as the prototype strain D388 (Appendix A). These strains demonstrated high nucleotide similarities ranging from 98.8% to 99.6% and amino acid similarities between 99.0% and 100%.

In contrast, the 2020 strain showed greater similarity to strains isolated in China (Fuzhou-20 strains) and a strain from Thailand, with nucleotide and amino acid similarities ranging from 98.1% to 99.2% and 98.1% to 99.0%, respectively (Table 1).

NSP4 gene

In the phylogenetic analysis of the NSP4 gene, known for its significant intrinsic variability, approximately thirty previously reported lineages were included. The study strains were classified into two major lineages: lineage VI and lineage XII. Nucleotide and amino acid similarities among the study strains ranged from 92.0% to 99.0% and 96.0% to 99.4%, respectively.

The strain isolated in 2020, RVA/Human-wt/BEN/20P093/2020/G3P[8], clustered within lineage VI, exhibiting nucleotide and amino acid similarities of 98.5% to 99.8% and 97.7% to 99.4%, respectively, with other lineage VI strains.

Notably, all strains assigned to lineage XII, including the Benin strains RVA/Human-wt/BEN/22011/2022/G3P[8] and RVA/Human-wt/BEN/22013/2022/G3P[8], predominantly exhibit a G8P[8] genetic background (Appendix A). The only exception is a Vietnamese strain isolated in 2019, which represents a DS-1-like G9P[8] reassortant that emerged during the G9P[8] dominance in Vietnam [50]. Nucleotide similarities for lineage XII strains ranged from 98.1% to 99.4%, with amino acid similarities ranging from 98.3% to 100%.

Additionally, a significant portion of the Benin T2 genogroup, including strains with G2P[4], G12P[6], and G3P[6] genotypes, were distributed across multiple lineages (VI, IX, XII, and XXIII). This extensive diversity of rotavirus strains circulating in Benin likely promotes the emergence of reassorted genotypes, as observed in this study.

NSP5 gene

Four lineages have been delineated for the NSP5 gene, with all study strains clustering within lineage IV. These strains demonstrated nucleotide and amino acid similarities of 99.7% to 100% and 100%, respectively. This lineage is consistent with strains identified globally, including those from Mozambique, Ghana, Kenya, and Malawi. The nucleotide similarities among these globally distributed strains ranged from 97.3% to 99.5%, while amino acid similarities ranged from 99.5% to 100% (Appendix A and Table 1).

### 3.3. Amino Acids Changes Within Antigenic Regions

Changes within G3 antigenic regions

Nine antigenic variable regions have been described for the VP7 gene of rotavirus [51]. A comparative analysis of these regions was conducted between the study strains, the equine-like prototype strain D388, the typical equine strain (RVA/Horse-wt/IND/Erv105/2003-2005/G3), and other non-equine-like strains previously identified in Benin and Mozambique. Using the reference strain RVA/Human-wt/AUS/D388/2013/G3P[8], the study strains displayed complete amino acid conservation across these antigenic regions. This suggests that the equine-like strains from Benin likely share the same antigenic characteristics as the prototype strain D388. Given the documented involvement of these strains in significant diarrheal outbreaks in various countries [23,24,52,53], their presence is concerning and underscores the need for real-time surveillance.

Outside of the variable regions, two amino acid substitutions—K250E and I260V—were observed across all three study strains. Additionally, the strain RVA/Human-wt/BEN/20P093/2020/G3P[8] exhibited a unique mutation, A156V (Appendix A).

Changes within P[8] antigenic regions

The VP4 protein, which undergoes trypsin cleavage to form the VP8 and VP5 subunits, has been instrumental in identifying key antigenic regions. VP8 contains five antigenic sites (8-1, 8-2, 8-3, 8-4, and 8-5), while VP5 comprises five regions (5-1, 5-2, 5-3, 5-4, and 5-5). Together, these two subunits encompass a total of 37 amino acid residues (25 in VP8 and 12 in VP5) [54].

Antigenic site comparison between the study strains, vaccine strains (Rotarix and Rotateq), the reference Wa-like strain, and two vaccine-derived strains revealed four amino acid substitutions: N195G, N195D, N113D, and D133N (Appendix A).

## 4. Discussion

This study reports, for the first time, the presence of equine-like G3P[8] rotavirus strains in Benin. Initially detected in East Asia (Japan, Indonesia) and Australia in 2013, these strains later spread to Europe, South America, and the USA. The study strains are closely related to the human-derived D388 strain, the prototype of equine-like G3P[8] (RVA/Human-wt/AUS/D388/2013/G3P[8]), isolated in Australia in 2013 [22,23,24,25,26,27,28]. The global circulation of equine-like G3P[8] has been attributed to reassortment between a VP7-G3 gene of unknown equine origin and a DS-1-like strain such as G2P[4] [32,33]. Other studies have also suggested reassortment between G3P[8] and DS-1-like strains, though primarily in non-equine-like G3 variants [50]. In this study, a comprehensive phylogenetic analysis and available rotavirus genomic data from Benin did not provide a clear explanation for the local reassortment process leading to these strains’ emergence. The limited genomic data in Africa, particularly in Sub-Saharan regions, further complicates efforts to trace their origins and transmission. To date, only two African countries, Kenya and Tanzania, have reported the circulation of equine-like G3P[8] strains [30,31]. In contrast, Kenya and Malawi have described classical G3P[8] strains with Wa-like backbones [31,52]. This raises the question of whether the absence of equine-like strains in previous reports reflects their actual non-circulation or merely a lack of genomic surveillance. Based on current evidence, the presence of equine-like strains in Benin likely results from sub-regional or continental importation rather than local reassortment, emphasizing the significant role of globalization, technological development, human migration, and international trade in pathogen spread across borders.

Given that G2P[4] strains are the second most prevalent genotype in Benin [7], local reassortment with G3P[8] might have been expected. However, no prior studies have reported G3P[8] strains in Benin. Instead, only G3P[6] strains with a DS-1-like backbone have been documented, clustering within G3-lineage I and distinctly separate from the study strains (Figure 1). These findings support the hypothesis of an introduction rather than a complex reassortment event. Expanding genomic surveillance in Africa is crucial for understanding the transmission dynamics of these strains and guiding public health interventions.

Most classical G3P[8] strains isolated in Africa and worldwide belong to lineage I [8,31,53], whereas a recent report from Mozambique identified G3-lineage-III strains [29]. In contrast, the strains in our study, unlike the previously reported classical G3P[8] strains, fall within lineage IX, a recently proposed classification described by Katz et al. (2019) [25]. These strains clustered within this lineage along with others originating from Kenya, Japan, Indonesia, the Dominican Republic, the USA, Thailand, Australia [23,25,28,31,54,55], and recently from Italy [56], as well as from Malaysia, where the Malaysian authors referred to it as Equine-like G3 lineage I [34].

Emerging evidence suggests that equine-like G3P[8] strains with a DS-1-like backbone can rapidly reassort with other genogroups, particularly Wa-like strains, which remain predominant in many countries. A study from Colombia reported an equine-like Wa-like G3P[8] strain resulting from reassortment between a previously unreported equine-like DS-1 variant and a local Wa-like strain, primarily G12P[8] [32]. Given the increasing emergence of G12P[8] in some African settings, such as Benin [7], especially in the post-vaccine introduction era [4,57,58], close monitoring is essential to detect early reassortment events. This is particularly important as such mutations could enhance pathogenicity and negatively impact child health [23,24].

The VP4-P[8] protein of the study strains was classified within lineage III, consistent with prior reports from Africa and beyond [29,34]. Structural and non-structural proteins (VP4, VP2, VP6, NSP3, NSP4) of the equine-like strains isolated in 2022 clustered with recent G8P[8] strains from the USA and Japan (2019), while the 2020 strain grouped with the prototype D388 strain and G3P[8] strains from the Dominican Republic, India, Japan, and, in some cases, Kenya, Rwanda, and Malawi for VP6. Across all VP and NSP genes, the study strains were classified into various sub-lineages and clades. Notably, the NSP4 gene in both pre- and post-vaccination strains from Benin was distributed across four different lineages, highlighting the genetic diversity of circulating strains. These findings emphasize the need for robust genomic surveillance to inform public health interventions. Since rotavirus surveillance in Benin has been limited to the southern region, expanding it nationwide would provide a more comprehensive dataset.

This study proposes a new NSP2 lineage (VI) based on identity/similarity cut-off values and phylogenetic analysis, showcasing the genetic variability of circulating strains and emphasizing the need for further genomic studies in Benin and the broader West African region. Benin introduced the ROTAVAC^®^ vaccine (G9P [11]) into its Expanded Program on Immunization (EPI) in 2019. Given the distinct genotypic composition of this vaccine compared to equine-like strains (G3P[8]), an antigenic region mutation analysis was not feasible. However, comparison of the VP7-G3 gene with the prototype D388 strain suggests that the study strains share antigenic properties with other equine-like strains. Reports associating equine-like G3P[8] with diarrhea outbreaks [23,24,34] highlight the need for close monitoring and genotype-clinical outcome correlation studies in children.

Although no variations were detected in the antigenic region, three mutations were identified: K250E and I260V in all three study strains, and A156V in RVA/Human-wt/BEN/20P093/2020/G3P[8]. Since these mutations occur outside the antigenic regions, they are less likely to directly affect antigenicity or the immune system’s recognition of the virus. I260V and A156V are classified as conservative mutations, meaning they are less likely to significantly alter the protein’s function. In contrast, K250E is considered a non-conservative mutation due to the substitution of a positively charged lysine with a negatively charged glutamic acid. This change could potentially impact the protein’s overall structure, stability, or interactions with other viral components, potentially influencing viral fitness or replication efficiency. However, it is less likely to directly affect immune recognition or vaccine effectiveness. Further investigation is required to elucidate the full implications of these mutations.

Mutations in the VP4-P[8] protein were also identified, including N195G, N195D, N113D, and D133N. These substitutions involve shifts from the polar, uncharged asparagine (N) to the polar, charged aspartic acid (D), as well as the unique glycine. Most of these mutations are semi-conservative, with probably minimal impact on the overall structure and function of the proteins. However, the non-conservative N195G substitution, in particular, warrants further investigation due to its potential implications. Indeed, glycine, due to its small size and lack of a side chain, provides significant flexibility to the protein backbone. This increased flexibility near position 195 could result in conformational changes within the VP4 protein, particularly in the VP8 subunit. These changes may alter the local structure, potentially influencing the antigenic properties of the protein. As a result, the immune system’s ability to recognize and mount an effective neutralizing response could be impaired, impacting the efficacy of immune detection. Such modifications could also affect vaccine efficacy, particularly for vaccines like Rotarix and RotaTeq, which target VP4. Alterations in the antigenic sites, like the N195G mutation, may reduce the effectiveness of the antibody response elicited by these vaccines. Additionally, the mutation may have functional consequences for viral entry, as VP4 is critical for binding to host cell receptors. Any change in receptor affinity or specificity could influence the virus’s ability to infect host cells, potentially altering its infectivity and tropism. Therefore, the N195G mutation may have far-reaching implications for viral transmission and disease outcomes. Nonetheless, these antigenic and functional implications remain speculative at this stage, and further experimental validation is needed to elucidate the true impact of the N195G mutation.

The main limitation of this study is its sample size. A larger genomic dataset from Benin would provide more precise insights into strain transmission dynamics and would allow for a more comprehensive phylodynamic study. Expanding surveillance beyond the southern region is essential to establish a comprehensive post-vaccine introduction map of circulating rotavirus strains. Further genomic studies are necessary to strengthen rotavirus surveillance and inform future vaccination strategies in Africa.

## 5. Conclusions

This report provides the first description of the equine-like G3P[8] strain with a DS-1-like backbone in Benin and West Africa, highlighting the need to include this globally emerging viral variant in routine RVA surveillance in Benin.

This study, at this stage, supports the hypothesis of an introduction of the strain into Benin rather than a local reassortment event, due to the limited availability of national and regional genomic data. While further comprehensive phylodynamic investigations are needed to trace the origin and transmission dynamics of these strains more precisely, we also emphasize the importance of expanding genomic surveillance efforts and ensuring open access to sequencing data, particularly in African countries.

Ultimately, we identify a novel NSP2 lineage with a DS-1-like backbone, designated NSP2-lineage VI.

## Figures and Tables

**Figure 1 viruses-17-01091-f001:**
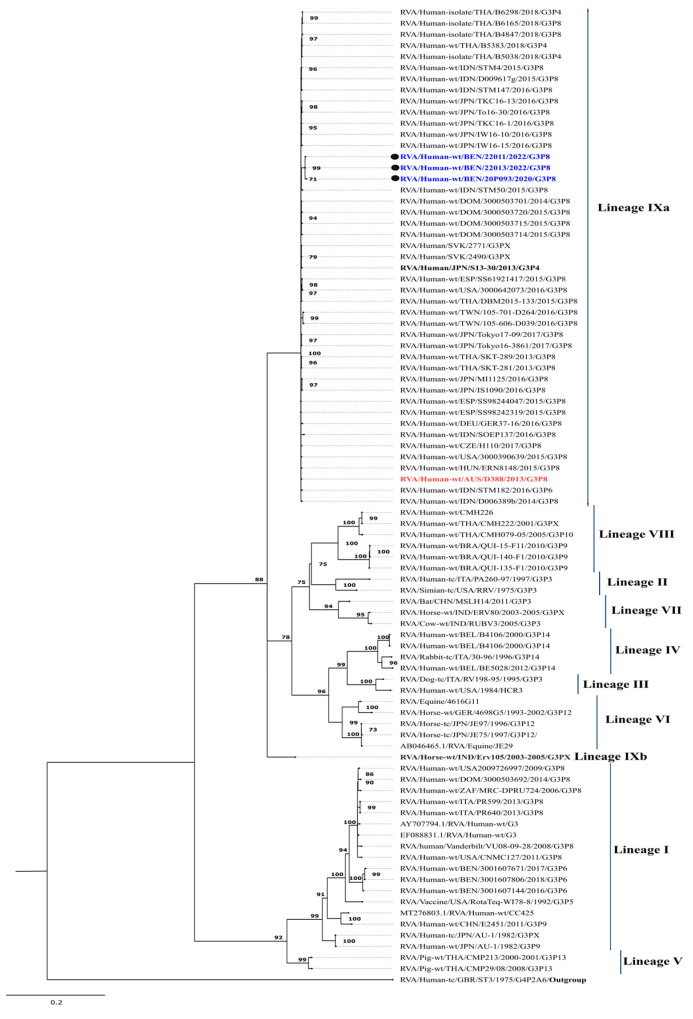
Maximum likelihood phylogenetic tree depicting the genetic relationships of the study strains with other globally circulating G3 strains retrieved from the GenBank database. The phylogenetic inference for the G3 gene was performed using the TPM1uf+G4 evolutionary model. Study strains are represented by black circles and are color-coded in blue. The human-derived D388 strain, the prototype of the equine-like G3P[8] strain, is color-coded in red. The equine strain Erv 105 is bolded. Only bootstrap values of ≥70% are shown at each branch node. The scale bar represents nucleotide substitutions per site.

**Figure 2 viruses-17-01091-f002:**
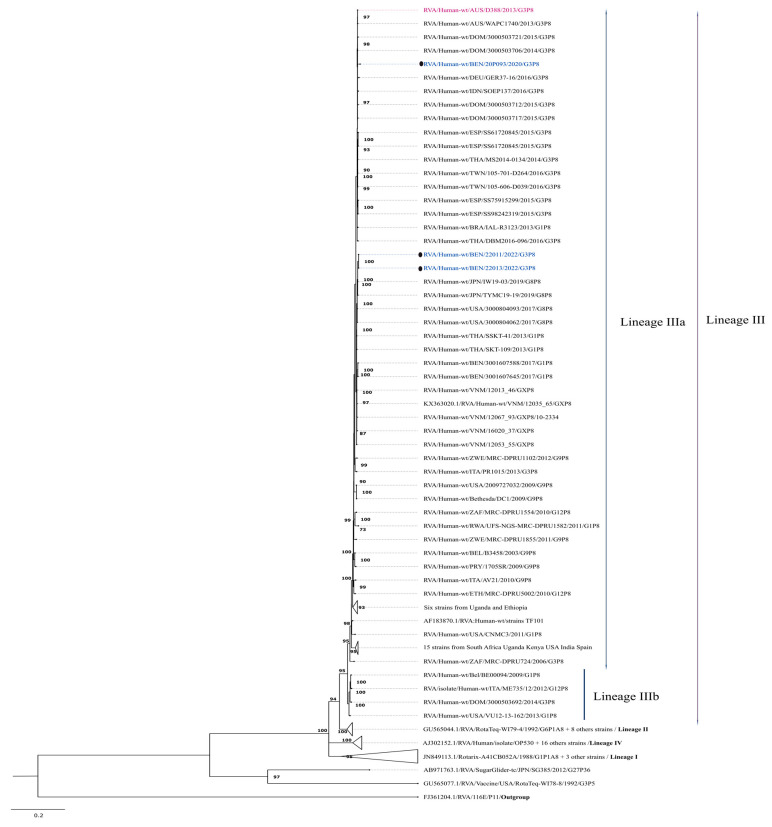
Maximum likelihood phylogenetic tree depicting the genetic relationships of the study strains with other globally circulating P[8] strains retrieved from GenBank. The phylogenetic inference for the P[8] gene was performed using the TIM3+I+G4 evolutionary model. Study strains are represented by black circles and are color-coded in blue. The human-derived D388 strain, the prototype of equine-like G3P[8] strains, is color-coded in red. The equine strain Erv 105 is in bold. Only bootstrap values of ≥70% are shown at each branch node. The scale bar represents nucleotide substitutions per site.

**Table 1 viruses-17-01091-t001:** Summary of nucleotide and amino acid identities of the different genes.

Genes/Genogroup	RVA Strains Compared	Shared Sequence Identity	
		Nucleotide Identity (%)	Amino Acid Identity (%)
VP7-G3	Among Benin G3 strains	99.6–100	99.7–100
	All G3-lineage IX	90.6–99.2	97.2–99.7
VP4-P[8]	Among Benin P[8] strains	97.7–100	97.7–100
	All P[8]-Lineage III	95.3–99.0	96.4–98.8
Genogroup I2	Among Benin VP6 strains	97.9–100	99.7–100
	All VP6-Lineage V	95.8–99.6	98.2–100
Genogroup R2	Among Benin VP1 strains	96.6–99.5	98.9–99.9
	All VP1-Lineage V	93.7–99.1	98.4–99.8
Genogroup C2	Among Benin VP2 strains	98.7–100	99.8–100
	All VP2-Lineage IV	95.7–99.7	99.1–100
Genogroup M2	Among Benin VP3 strains	99.5–100	99.5–100
	All VP3-Lineage VII	95.9–99.1	97.7–99.6
Genogroup A2	Among Benin NSP1 strains	95.8–99.3	96.3–99.6
	All NSP1-Lineage IV	93.8–99.2	93.3–99.2
Genogroup N2	Among Benin NSP2 strains	86.6–100	95.0–100
	All NSP2-Lineage V	80.8–99.1	94.3–99.7
	All Benin strains and NSP2-Lineage VI (new)	86.6–100	95.3–100
	Among lineage VI	98.6–100	99.7–100
	Lineage VI and lineage I (HQ650123/RVA/Human-tc/USA/DS-1/1976/G2P4)	88.1	94.6
	Lineage VI and lineage II	84.5–85.3	92.1–93.1
	Lineage VI and lineage III	87.8	94.3–94.6
	Lineage VI and lineage IV	88.0	94.6
	Lineage VI and lineage V	86.4–87.1	94.3–95.6
Genogroup T2	Among Benin NSP3 strains	96.2–99.8	97.1–100
	All NSP3-Lineage V	98.8–99.6	99.0–100
Genogroup E2	Among Benin NSP4 strains	92.0–99.0	96.0–99.4
	All NSP4-Lineage VI	98.5–99.8	97.7–99.4
	All NSP4-Lineage XII	98.1–99.4	98.3–100
Genogroup H2	Among Benin NSP5 strains	99.7–100	100
	All NSP5-Lineage IV	97.3–99.5	99.5–100

## Data Availability

The raw data supporting the conclusions of this article will be made available by the authors on request. The nucleotide sequences have been submitted to GenBank and are currently being assigned accession numbers.

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
