# Peer review of "Unveiling a Shift in the Rotavirus Strains in Benin: Emergence of Reassortment Intergenogroup and Equine-like G3P[8] Strains in the Post-Vaccination Era"

_viruses, 2025, doi:10.3390/v17081091_

Round 1

Reviewer 1 Report

Comments and Suggestions for Authors

In this manuscript the authors describe for the first time, the molecular chracterization in terms of it genome constellation  of several equine-like G3P[8] rotavirus strains with a DS- 551 1-like backbone detected in Benin and West Africa

Samples were collected pre and post vaccination with a G9P[11] RVA vaccine.

The  strains under study are  closely related to the human-derived D388 strain, the prototype of equine-like G3P[8] (RVA/Human-wt/AUS/D388/2013/G3P[8]), isolated in Australia in 2013.

The analyzed data is relevant, adding epidemiological information from Africa, where Kenya  and Tanzania, have reported the circulation of equine-like G3P[8] strains and now Benin represent a third country reporting this RVA strain.

Regarding the origin of the strain, I wonder if the author can do  a phylodynamic study with strains from other countries is order to add some to discussion to this issue. 

The MS is well written and analyzed, overall I find it suitable for publication in it present form or with the suggested photodynamic study.

Author Response

Comment 1:

Regarding the origin of the strain, I wonder if the author can do a phylodynamic study with strains from other countries is order to add some to discussion to this issue. The MS is well written and analyzed, overall I find it suitable for publication in it present form or with the suggested photodynamic study.

Response 1:

Thank you for pointing this out. We fully agree that phylodynamic analysis offers valuable opportunities to deepen our understanding of the transmission dynamics and evolutionary patterns of emerging rotavirus strains. Although a formal phylodynamic analysis using Bayesian methods such as BEAST was not performed in this study, we conducted an extensive phylogenetic analysis that incorporates both temporal and geographical information, as indicated in the nomenclature of the strains included in the analysis. This approach enabled us to explore the evolutionary relationships of the study strains (the equine-like G3P[8]), in the context of global rotavirus diversity, using publicly available sequence data from NCBI.

To strengthen the relevance and representativeness of our comparative framework, we carefully curated a diverse set of reference strains from both recent and earlier studies in the field. This dataset ensured broad temporal and geographical coverage, enhancing the robustness of our phylogenetic analysis and interpretations. Furthermore, our whole-genome sequencing data provided meaningful baseline insights into the genetic makeup and evolutionary history of the study strains. Notably, for the VP7-gene, the equine-like G3P[8] strains detected in Benin showed strong genetic relatedness to the human-derived D388 strain, the prototype of this lineage (RVA/Human-wt/AUS/D388/2013/G3P[8]), which was first identified in Australia in 2013.

We believe that this level of analysis already provides substantial insight into the genetic diversity and circulation patterns of the rotavirus strains detected in the post-vaccine era in Benin. Therefore, we have not made additional changes to the core analysis in the manuscript. However, we now acknowledge the absence of a formal phylodynamic study as a limitation in the Discussion section (Page 14, Paragraph 2, Lines 568–569) by updating the sentence as follows (see "Track Changes" in the manuscript):

« The main limitation of this study is its sample size. A larger genomic dataset from Benin would provide more precise insights into strain transmission dynamics and would allow for a more comprehensive phylodynamic study ».

We also acknowledge the potential added value of such an approach and plan to incorporate phylodynamic tools in future studies as more longitudinal and representative sequence data become available.
Thanks once again.

Reviewer 2 Report

Comments and Suggestions for Authors

The manuscript titled “Unveiling shift in the rotavirus strains in Benin: Emergence of reassortment intergenogroup and equine-like G3P[8] strains in the post-vaccination era” provides valuable data on the emergence and genetic evolution of equine-like G3P[8] strains in Benin following vaccine introduction. Although the appearance of such strains has been well documented globally, limited data are available from Africa due to the lack of comprehensive genomic surveillance.

The detection of equine-like G3P[8] strains, along with evidence of intergenogroup reassortment and the identification of a novel NSP2 lineage, is well supported by extensive genomic and phylogenetic analyses.

The manuscript is generally well-structured, with a logical progression from the introduction to the discussion. Figures and phylogenetic trees are clear and informative, although some supplementary figures (e.g., S1–S2) that are referenced in the text should be integrated into or summarized more explicitly in the main manuscript.

Minor Revisions and Suggestions:

  • Details regarding sample selection criteria and metadata (e.g., patient age, symptoms, and clinical severity) are lacking. Including this information would enhance the epidemiological relevance of the study. How many samples were tested for rotavirus infections? How were the samples selected for phylogenetic analysis? Were preliminary VP7 and VP4 analyses performed?

  • On page 5, lines 173–174: This content should be moved to the “Sampling Method” section.

  • Table 1 should be moved to the Supplementary Data section, while Table S1 should be integrated into the main text.

  • The nomenclature of rotavirus strains should be standardized across all phylogenetic trees.

  • While the novelty of the study is commendable, the mechanistic explanation for the reassortment event remains speculative due to limited regional genomic data. This limitation is acknowledged in the text but should be more explicitly emphasized in the conclusions.

  • Sequences of Italian G3P[8] strains, which have been recently detected, should be included in the phylogenetic analyses if available.

  • Hypotheses regarding vaccine pressure and its potential impact on vaccine effectiveness should be presented with greater caution.

  • The antigenic implications of VP4 mutations (especially N195G) are potentially significant but remain speculative. Emphasize the need for experimental validation to support these claims.

Comments on the Quality of English Language

The English is mostly clear and readable; however, several sections would benefit from language polishing to improve fluency and readability. A professional language edit is recommended to ensure smoother syntax and grammar.

Author Response

Comment 1:

Details regarding sample selection criteria and metadata (e.g., patient age, symptoms, and clinical severity) are lacking. Including this information would enhance the epidemiological relevance of the study. How many samples were tested for rotavirus infections? How were the samples selected for phylogenetic analysis? Were preliminary VP7 and VP4 analyses performed?

Response 1:

Thank you for pointing this out. We agree that providing additional epidemiological details strengthens the relevance of the study. Accordingly, we have updated the Materials and Methods section (Page 3, Paragraph 2, Lines 120–134) to include the following information:

"The first sample, identified as RVA/Human-wt/BEN/20P093/2020/G3P[8], was obtained from a 5-month-old male infant residing in Sémè-Kpodji. The patient presented with a 5-day history of diarrhea without vomiting and exhibited a relatively mild clinical presentation (no signs of lethargy or unconsciousness, no sunken eyes, and normal drinking behavior), although moderate dehydration was noted. The second sample, RVA/Human-wt/BEN/22011/2022/G3P[8], was collected from an 8-month-old male infant from Cotonou. The child presented with fever (38.3°C), diarrhea and vomiting (4 episodes per day), without lethargy or unconsciousness, and maintained normal drinking behavior. However, sunken eyes and moderate dehydration were observed. The third sample, RVA/Human-wt/BEN/22013/2022/G3P[8], was derived from a 9-month-old male infant, also from Sémè-Kpodji. This patient presented with diarrhea and vomiting (4 episodes per day), accompanied by lethargy/unconsciousness, sunken eyes, poor drinking behavior, and moderate dehydration, but no fever was reported.

All three children had received the ROTAVAC® vaccine as part of the national immunization program.

As mentioned in the manuscript, all samples included in the study were collected as part of the national rotavirus surveillance program in Benin. Initial screening for rotavirus infection was performed using an ELISA-based method. Under a Material Transfer Agreement (MTA) and collaboration framework, a total of 89 rotavirus-positive stool samples were transferred to the University of the Free State–Next Generation Sequencing (UFS-NGS) Unit, a WHO Collaborating Centre for Vaccine Preventable Diseases (VPD) Surveillance and Pathogen Genomics in South Africa, for whole-genome sequencing.

Whole-genome sequencing was performed on all 89 samples using the Illumina MiSeq platform. Among these, three samples were identified as equine-like G3P[8] strains which is the focus of the present study. These samples were therefore selected for full-genome characterization and phylogenetic analysis. Preliminary VP7 and VP4 genotyping was not performed prior to whole-genome sequencing. Instead, ELISA-positive samples were directly processed using the TRIzol extraction method, which included a specific enrichment step for double-stranded RNA to optimize rotavirus genome recovery. The quality and integrity of the extracted nucleic acids were assessed and confirmed prior to downstream processing, including cDNA synthesis, library preparation, and sequencing.

Comment 2:

On page 5, lines 173–174: This content should be moved to the “Sampling Method” section.

Response 2:

Thank you for this helpful suggestion. As requested, the content originally found on Page 5, Lines 173-174 has been moved to the "Sampling Method" section.

Comment 3:

Table 1 should be moved to the Supplementary Data section, while Table S1 should be integrated into the main text.

Response 3:

Thank you for the suggestion. In accordance with your recommendation, Table S1 has been integrated into the main manuscript, and the former Table 1 has been moved to the Supplementary Materials and is now labeled as Table S1.

Comment 4:

The nomenclature of rotavirus strains should be standardized across all phylogenetic trees.

Response 4:

Thank you for this valuable suggestion. We fully agree with the importance of consistent strain nomenclature for clarity and comparability. Accordingly, we have reviewed all phylogenetic trees and standardized the rotavirus strain names to ensure consistency, to the best extent possible. In line with standard practices in the field, we adopted the nomenclature format RVA/Host-species/Country/StrainID/Year/GxP[x] for most strains. However, for certain cases, particularly where strains were difficult to retrieve from GenBank due to incomplete or inconsistent strain identifiers, we included the corresponding GenBank accession numbers to facilitate accurate referencing. Additionally, for reference strains representing specific lineages, accession numbers were also added. This was done to facilitate accurate and straightforward identification of these sequences by other researchers.

Comment 5:

While the novelty of the study is commendable, the mechanistic explanation for the reassortment event remains speculative due to limited regional genomic data. This limitation is acknowledged in the text but should be more explicitly emphasized in the conclusions.

Response 5:

Thank you for this valuable observation and suggestion. As recommended, we have revised the conclusion to more explicitly emphasize this limitation. The updated conclusion now reads as follows:

This report provides the first description of the equine-like G3P[8] strain with a DS-1-like backbone in Benin and West Africa, highlighting the need to include this globally emerging viral variant in routine RVA surveillance in Benin. This study, at this stage, supports the hypothesis of an introduction of the strains into Benin rather than a local reassortment event, due to the limited availability of national and regional genomic data.

While further comprehensive phylodynamic investigations are needed to more precisely trace the origin and transmission dynamics of these strains, we also emphasize the importance of expanding genomic surveillance efforts and ensuring open access to sequencing data, particularly in African countries.

Ultimately, we identify a novel NSP2 lineage with a DS-1-like backbone, designated NSP2-lineage VI.

Comment 6:

Sequences of Italian G3P[8] strains, which have been recently detected, should be included in the phylogenetic analyses if available.

Response 6:

Thank you for this insightful suggestion. In response, we have incorporated available Italian G3P[8] strains into our dataset across all 11 rotavirus gene segments. We used both the NCBI and ViPR databases and optimized our BLAST search parameters to ensure the inclusion of recent, relevant, and closely related strains.

However, we were unable to include certain strains referenced in recent publications due to the lack of public availability or restricted access to non–open access articles. While we aimed to be as comprehensive as possible, we also placed particular emphasis on including strains from African countries to better contextualize the evolutionary relationships of the study strains.

We are currently pursuing follow-up studies and will ensure the integration of additional Italian strains as more sequences become publicly available.

Comment 7:

Hypotheses regarding vaccine pressure and its potential impact on vaccine effectiveness should be presented with greater caution.

Response 7:

Thank you for this valuable suggestion. We fully agree with the importance of presenting any hypotheses related to vaccine pressure and its potential impact on vaccine effectiveness with appropriate caution.

We have been careful not to make any claims unsupported by scientific evidence and have followed a rigorous methodological approach in our analysis and interpretation.

We are also aware of the limitations of our study, particularly the relatively small sample size and the limited post-vaccination time window, which may constrain the ability to draw some conclusions.This is a very important point, and we sincerely appreciate your attention to it.

Comment 8:

The antigenic implications of VP4 mutations (especially N195G) are potentially significant but remain speculative. Emphasize the need for experimental validation to support these claims.

Response 8:

Thank you for this thoughtful and constructive comment. We fully agree that the proposed implications of the N195G mutation in VP4 remain speculative and require experimental validation to be confirmed. To address this, we have revised the discussion to clearly acknowledge the hypothetical nature of these interpretations and to emphasize the need for further validation.The following sentence has been added at the end of the relevant paragraph in the Discussion section (Page 16, Paragraph 1, Lines 564-565):

 "Nonetheless, these antigenic and functional implications remain speculative at this stage, and further experimental validation is needed to elucidate the true impact of the N195G mutation."